# Raman Microspectroscopic Investigation and Classification of Breast Cancer Pathological Characteristics

**DOI:** 10.3390/molecules26040921

**Published:** 2021-02-09

**Authors:** Heping Li, Tian Ning, Fan Yu, Yishen Chen, Baoping Zhang, Shuang Wang

**Affiliations:** State Key Laboratory of Photon-Technology in Western China Energy, Institute of Photonics and Photon-Technology, Northwest University, Xi’an 710127, China; 18291861688@163.com (H.L.); ntoptics@163.com (T.N.); 15771915828@163.com (F.Y.); awawerwer2007@163.com (Y.C.); zbp18438591162@163.com (B.Z.)

**Keywords:** breast tissues, raman spectroscopy, PCA–LDA, diagnosis

## Abstract

Breast cancer is one of the major cancers of women in the world. Despite significant progress in its treatment, an early diagnosis can effectively reduce its incidence rate and mortality. To improve the reliability of Raman-based tumor detection and analysis methods, we conducted an ex vivo study to unveil the compositional features of healthy control (HC), solid papillary carcinoma (SPC), mucinous carcinoma (MC), ductal carcinoma in situ (DCIS), and invasive ductal carcinoma (IDC) tissue samples. Following the identification of biological variations occurring as a result of cancer invasion, principal component analysis followed by linear discriminate analysis (PCA-LDA) algorithm were adopted to distinguish spectral variations among different breast tissue groups. The achieved results confirmed that after training, the constructed classification model combined with the leave-one-out cross-validation (LOOCV) method was able to distinguish the different breast tissue types with 100% overall accuracy. The present study demonstrates that Raman spectroscopy combined with multivariate analysis technology has considerable potential for improving the efficiency and performance of breast cancer diagnosis.

## 1. Introduction

Among women cancer patients within the age range of 33–55 years, breast cancer is the second leading cause of cancer death [1,2]. It was estimated that there would be 276,480 new breast cancer cases (30% of all new cancer cases) and 42,170 breast cancer-related deaths (15% of all cancer deaths) by 2020, making breast cancer the main of all new cancer cases [3]. However, statistics show that if breast cancer is treated at an early stage, about 90% of patients can be cured [4]. The gold standard for breast cancer diagnosis usually includes fine-needle aspiration cytology, core needle biopsy, and surgical resection with subsequent histopathological analysis of the resected breast tissue. Although advances in screening technology can allow early diagnosis of breast cancer in many cases, the significant number of diagnoses that eventually lead to death (~20% at 15 years) is the primary impetus for improving surgical interventions [5,6].

Breast conserving surgery (BCS) is a common method in the treatment of breast cancer. The purpose of BCS is to remove the entire tumor while preserving as much healthy breast tissue as possible. However, this is a challenge because of the lack of tools to assess complete resection of marginal tumors during surgery. Unfortunately, in >30% of breast tumor excisions, the surgeon inadvertently cuts into tumor tissue and leaves cancer behind [7,8]. These positive surgical margins following lumpectomy are well-documented risk factors for local recurrence and disease-specific mortality. However, it usually takes 1–2 weeks to evaluate the edge of the operation, and if the margin is positive, the patient needs to have a second operation, which causes discomfort and increases the economic burden [9]. Therefore, additional techniques are needed to assess the margin of BCS specimens within the intraoperative time frame (i.e., less than 30 min). Histopathological evaluation of frozen sections and cytological imprint preparation (i.e., touch preparation) analysis can evaluate the edge during this period [10,11], but due to the large size of the samples from BCS, sampling errors [10,12], sample preparation artifacts [13], as well as the time–cost factor for pathologists, these analyses are generally considered impractical. Therefore, there is a practical need to develop a novel technique that efficiently discriminates between healthy and tumor tissue through a complete and precise understanding of the molecular pathways linked to breast cancer progression to improve the efficiency and precision of surgical treatments.

Raman spectroscopy (RS) is a nondestructive approach providing detailed quantitative biochemical information of biological samples thanks to the simple detection of the vibrational properties of molecules by light scattering, without using external labels or dyes [14]. It is especially suitable for in vivo measurements because the power and excitation wavelength used are non-destructive to tissues and have relatively large penetration depth [15]. The integration of confocal microscopy to Raman systems has allowed the analysis of samples with small volumes, in particular the depth profiling of biological samples. After years of development, clinicians and researcher communities believe that RS is promising for the early diagnosis of cancer, the identification of cancer progression, and intraoperative guidance [16,17,18,19,20], especially for breast cancer [21,22], even though it is still necessary to build a correlation between the collected spectral information and tissue pathological characteristics by using multivariate analysis algorithms for helping clinicians to make precise diagnosis [23].

In the present research, we aimed to characterize the spectral features of fresh frozen sections in healthy control (HC), solid papillary carcinoma (SPC), mucinous carcinoma (MC), ductal carcinoma in situ (DCIS), and invasive ductal carcinoma (IDC) specimens, so to identify the spectral characteristics associated with cancer progression and to facilitate the development of Raman-based tumor detection algorithms. Multivariate analysis models, i.e., principal component analysis (PCA) followed by linear discriminate analysis (LDA), were further utilized to analyze and classify the Raman spectra of the different types of tissue. This study demonstrates that Raman spectroscopy combined with multivariate data analysis can be used for early diagnosis of breast cancer.

## 2. Results and Discussion

### 2.1. Pathological Analysis

Significant morphological differences were observed between HC, SPC, MC, DCIS, and IDC tissue samples, (Figure 1). In the process of cancer cell infiltration, a large number of nuclei can be detected because of their purple hematoxylin and eosin (H&E) appearance. In Figure 1B, the stained image of a solid papillary carcinoma sample shows spindle cell morphology, while a mucinous carcinoma sample displayed mild nuclear atypia and abundant mucus, as shown in Figure 1C. Compared with healthy breast tissue shown in Figure 1A, cancer cells in DCIS tissue were distributed around the duct without breaking through the basement membrane, as shown in Figure 1D. Conversely, in an IDC sample (Figure 1E), cancer cells did break through the basement membrane of a breast duct, exhibiting an anisotropic distribution pattern. The detailed information of the sample sources is presented in Table 1.

### 2.2. Raman Spectral Analysis

Raman spectra were acquired using an Alpha 500R confocal Raman microscopy system (WITec GmbH, Ulm, Germany) coupled with a helium–neon (He–Ne) continuous 633 nm laser (35 mW at 633 nm, Research Electro-Optics, Inc., Boulder, CO, USA). Each spectrum was recorded over a period of 1.5 s. The tissue area of spectrum collection was determined by pathology experts, and 100 single spectra were collected for each tissue type. The spectra of each tissue type were averaged and then normalized for comparison.

The normalized spectra of healthy and cancerous breast tissues are shown in Figure 2. The main characteristic peaks were observed in both the HC samples and the cancerous (DCIS, MC) samples at the following locations (with their respective tentative biochemical assignments shown in parentheses): 868 cm^−1^ (C–C stretching, collagen) [24]; 1076 cm^−1^ (C–C stretching, lipid) [25]; 1267 cm^−1^ (lipids) [26]; 1302 cm^−1^ (CH_2_ twisting and wagging, phospholipids) [27]; 1440 cm^−1^ (CH_2_ and CH_3_ deformation vibrations, lipids) [28]; 1654 cm^−1^ (C=C lipid stretching) [29]; 2854 cm^−1^ (CH_2_ symmetric stretch, lipids) [30]; 2890 cm^−1^ (CH_2_ asymmetric stretch, lipids and proteins) [30]; 2934 cm^−1^ (CH_2_ anti-symmetric stretching, lipids) [31]. Compared with the peaks in HC and DCIS samples, the peak of lipids in MC tissue shifted from 2854 cm^−1^ to 2860 cm^−1^ (CH_2_ symmetric stretch, lipids). Meanwhile, some additional characteristic peaks were observed in IDC specimens, such as those at 754 cm^−1^ (symmetric ring breathing in tryptophan, protein) [31]; 1552 cm^−1^ (C=C stretching in tryptophan) [31]; 1608 cm^−1^ (C=C stretching in phenylalanine) [32]. Moreover, the IDC and SPC samples displayed some common peaks at 1250 cm^−1^ (amide III) [33] and 2885 cm^−1^ (CH_3_, lipids) [24], while the peak at 2854 cm^−1^ was not detectable.

Compared with cancerous tissue, the intensity of lipid-associated peaks at 1267, 1302, 1440, 2854, and 2890 cm^−1^ in the HC group was higher, which may be related to a high rate of cell division and the thinning of lipidic cell membranes in the process of cancer cell invasion and migration [34]. In addition, lipid peroxidation by reactive oxygen species (superoxide anion radicals, O^−2^) or iron complexes in cancer tissues may also reduce the lipid content [35]. The characteristic peaks of protein at 754, 1250, 1552, and 1608 cm^−1^ for the cancer samples had higher intensity than the corresponding ones recorded for the HC specimens. This indicated that protein levels were higher in cancerous tissue samples, possibly due to the large quantity of protein synthesized by cancer cells during uncontrolled growth [34].

In order to better understand the biological variations after cancer progression, one-way-ANOVA and Tukey’s honestly significant difference (HSD) post-hoc multiple tests were sequentially performed to depict the relative intensity of individual peaks in IDC, DCIS, SPC, MC samples; HC samples were used for comparison, as shown in Figure 3. Asterisks indicate the levels of significance. Remarkable variations can be observed in the spectral contribution of collagen (868 cm^−1^), lipids (1267 cm^−1^, 1302 cm^−1^, 1440 cm^−1^, 2890 cm^−1^), protein (1608 cm^−1^). The results showed that the intensity of the protein peaks in cancerous tissues were higher than that in HC samples, whereas the intensity of the lipid peaks was lower in cancer tissues. This means that during cancer invasion, the level of lipids is decreased, while that of proteins is increased. This may be due to the fact that tumors share a common phenotype of uncontrolled cell proliferation and, therefore, they consume large amounts of lipids and synthesize large amounts of proteins. It is worth noting that the intensity of the peak at 868 cm^−1^ (collagen) was higher for the HC group than for the IDC, DCIS, and SPC groups, possibly due to changes in protein structure during breast tissue carcinogenesis. These results are consistent with the single spectral analysis in Figure 2.

### 2.3. PCA–LDA Analysis

As shown in Figure 1, the spectral differences among different tissue types were usually very small, with obvious spectral overlaps and non-evident intensity changes. Therefore, we conducted a multivariate analysis including all recorded biochemical variations to distinguish the different types of breast tissues under investigation.

The Raman spectra of the five tissue types related to the low-wavenumber region (600–1800 cm^−1^) and the high-wavenumber region (2800–3000 cm^−1^) were considered and categorized by PCA to obtain corresponding PC scores and loading values. The first PC accounted for the largest variance within the spectral dataset (PC1, 74.9%), while PC2 and PC3 represented 9.4% and 7% of the total variance, respectively. The scatter plot of PC1 and PC2 shown in Figure 4A revealed a clear separation between cancerous tissues and healthy breast tissue. Basically, the spectra on the positive axis of PC1 chiefly belonged to cancerous tissue, whereas spectra of the healthy tissue were mostly located on the negative axis. It appeared that the zero line of PC1 could mainly be used to distinguish healthy tissue from cancer tissue. Similarly, the scatter plots of PC1 and PC3, as well as of PC2 and PC3 could also be used to distinguish different types of breast tissue, as shown in Figure 4B,C. The spectra on the positive axis of PC2 mainly belonged to the SPC and MC samples, while the spectra of the HC, DCIS, and IDC samples were mainly located on the negative axis. Most of the spectra of the DCIS group were located on the positive axis of PC3, while most of the spectra of the IDC group were distributed on the negative axis.

Moreover, we found that PCA not only was able to distinguish the spectra of different tissues, but also had the potential to derive molecular feature information related to tissue classification, depending on the corresponding loading spectra of each PC [36]. A clear spectral variability was observed, and moreover the loadings could be compared to pristine Raman spectra. The loadings of PC1, PC2, and PC3 are shown in Figure 5. The loading of PC1 was basically below the zero line, with obvious peak positions in the loading at 1267, 2860, 2900 cm^−1^, which can be attributed to the characteristic bands of lipid components. Compared with the single spectrum of Figure 2, the features of the PC1 loading appeared extremely similar to the spectral characteristics of healthy breast tissue, suggesting that PC1 can be used primarily to differentiate healthy tissue from cancer tissue. This is consistent with the distribution of the spectra of healthy breast tissue shown in the scatter plot of Figure 4A,B. The positive peaks in the spectral loading of PC2 were attributed to biochemical components such as protein at 868 and 1250 cm^−1^, phenylalanine at 1002 cm^−1^, lipids at 1302, 1450, 1662, and 2940 cm^−1^, and carotenoid at 1524 cm^−1^ [37], while the negative peaks were attributed to lipids at 2854 cm^−1^. The loading of PC2 with the characteristic spectrum of cancerous tissue (MC) revealed marked similarities between the two groups of spectra. These observations indicate that the features of PC2 extraction were derived mainly from the MC samples.

The PC3 loading was distributed on both sides of the zero line, with the most diagnostically significant features located in the positive region at 868, 1002, 1450, 1662, 2860, and 2900 cm^−1^ and in the negative region at 754 cm^−1^ (protein) [31] and 2934 cm^−1^. These peaks indicated that the positive characteristics of PC3 loading were mainly contributed by protein and lipid components. Combining the positive and the negative score distributions of PC3, it was revealed that the cancerous IDC and SPC tissues contained more proteins and lipids. Moreover, this also showed that PCA loading can suggest underlying biochemical differences between different types of breast tissue. In addition to the correlation between PCA loadings and spectral distribution in the score scatter plot, the results of the PCA analysis could be used to classify different biochemical components by comparing the collected spectra with a reference spectrum. Therefore, PCA can effectively analyze the biochemical differences between different samples using the spectral datasets we constructed.

The first three PC scores were used as input variables for LDA to establish a final diagnostic model. Figure 6 demonstrates the linear discriminant scores of all obtained spectral data in both fingerprint (600–1800 cm^−1^) and high-wavenumber range (2800–3000 cm^−1^) from the investigated samples obtained by the PCA–LDA algorithm. The scatter plot of the LDA discriminant scores shows that the spectra of MC tissue were mainly distributed on the negative axis of the first discriminant function, while the spectra of breast tissue from other experimental groups were mainly distributed on the positive axis. The spectra of DCIS and IDC were mainly on the positive axis of the second linear discriminant score, whereas the spectra of the HC group were mainly observed on the negative axis, and those of the SPC samples were evenly distributed on both sides of the zero line.

In the leave-one-out cross-validation (LOOCV) confusion matrix (Table 2), the training set was obtained by using the classification model based on the PCA–LDA algorithm combined with the LOOCV method. It can be seen that a partial misclassified spectrum of the HC, IDC, DCIS and SPC groups was obtained, possibly due to similar spectral features of these tissues. The overall accuracy of the cross-validation was 98.75%. In Table 3, the overall accuracy of the PCA–LDA model on the test set was 100%.

## 3. Experimental Section

### 3.1. Sample Preparation

A total of 12 healthy breast samples, from four female patients, was purchased from Alenabio (Xi’an, Shaanxi, China); the biopsies were performed using protocols approved by the IRB (Institutional Review Board) and the HIPAA (Health Insurance Portability and Accountability Act). Additionally approval was onbtained for commercial product development. SPC (*n* = 3), MC (*n* = 3), IDC (*n* = 6), and DCIS (*n* = 6) samples from 18 female patients with an average age of 50 years were obtained from clinical breast-conserving surgery performed in the Department of Breast Surgery, the First Affiliated Hospital of Xi’an Jiaotong University, Xi’an, China. This was a retrospective study, for which formal consent was not required, and the ethics practice is under the guidance of the Ethics Committee of the First Affiliated Hospital of Xi’an Jiaotong University (Xi’an, China). The lesion type was verified by the pathologist in the First Affiliated Hospital of Xi’an Jiaotong University. All research procedures, including sample collection, tissue section preparation, and spectral analysis, complied with current laws in China.

Immediately after lesion excision, the samples were embedded in optimal cutting temperature medium (OCT, Surgipath^®^ FSC 22^®^, Leica Biosystems, Buffalo Grove, IL, USA) and frozen in liquid nitrogen for a better preservation of their native morphology. Longitudinal sections of 12 μm of thickness were placed on gold-coated glass substrates (BioGold^®^ 63479-AS, Electron Microscopy Sciences, Philadelphia, PA, USA) for spectroscopic analysis, so to eliminate the background fluorescence from the microscope slides and optics for spectroscopic measurement [38,39]. Consecutive 5 μm thick sections were stained with hematoxylin and eosin (H&E) to facilitate a comparison of the spectral measurements with the histopathological results. Frozen sections were maintained in an acetone cooling bath for dehydration and stored at −20°C until transportation to Northwest University, Xi’an, China, for spectroscopic studies. Tissue sections were thawed for less than 30 min at room temperature prior to spectroscopic analysis or additional histological processing. Although the OCT medium would not comtaminate the tissue spectra [40], some drops of phosphate-buffered saline were used to wash the tissue sections to eliminate the OCT as much as possible.

### 3.2. Spectroscopic Acquisition

The equipment used for Raman spectroscopy has been described in detail previously [41,42]. Briefly, a single spectrum was collected using a WITec Alpha 500 confocal micro-Raman spectroscopy system (WITec GmbH, Ulm, Germany) using a 633 nm He–Ne laser source (35 mW, Research Electro-Optics, Inc., Boulder, CO, USA). A 100× microscope objective (NA = 1.25, EC Epiplan-Neofluar, Zeiss, Oberkochen, Germany) was used for spectral excitation and measurement. In total, 100 spectra were randomly acquired from each sample (HC, DCIS, SPC, MC, and IDC). Each spectrum was recorded over a period of 1.5 s using a Raman spectrometer (UHTS300, WITec GmbH, Ulm, Germany) incorporating a 600 mm^−1^ grating with a back-illuminated deep-depletion charge-coupled device camera (Du401A-BR-DD-352, Andor Technology, Abingdon, UK) at a resolution of approximately 3 cm^−1^.

### 3.3. Data Pre-Processing and Analysis

WITec Project FOUR software (WITec GmbH, Ulm, Germany) was used to preprocess all datasets that were obtained for band range selection, cosmic ray removal, background subtraction, and spectral smoothing, using the same parameters in each case. All Raman spectra were normalized using an area-under-the-curve method over the ranges 600–1800 cm^−1^ and 2800–3000 cm^−1^ to minimize the effects of sample and instrument variability.

The spectral datasets were mean-centered, then used to conduct additional analysis. PCA was used to simplify complexity and identify key variables in the multidimensional datasets [43]. As a supervised classification method, LDA can be used in combination with PCA to improve the performance of classification. The initial dataset is divided into a training set (80%) and a test set (20%) according to a certain proportion. The training set is used to build a classification model and optimize model parameters (number of principal components) by cross-validation, whereas the test set is used to evaluate the model classification performance. A leave-one-out cross-validation (LOOCV) technique was used to verify the performance of the diagnostic model based on the PCA–LDA algorithm for the classification of different tissue types.

## 4. Conclusions

There are still many challenges in using Raman spectroscopy to analyze biological samples. Biological samples (tissues, cells, and body fluids) exhibit weak spontaneous Raman signals, which require long detection time and can easily be subjected to interference by fluorescence. Therefore, when introducing high signal-to-noise Raman spectroscopy as a diagnostic tool, issues associated with interference of background fluorescence, artifacts, and weak signals should be overcome. Meanwhile, a diagnostic model based on multivariate statistical analysis is also required to help clinicians make precise diagnoses.

In this context, this study explains the significant biochemical differences between healthy tissue and cancerous breast tissue. The biochemical changes in different types breast cancer tissues can be attributed to the uncontrolled proliferation of cells during carcinogenesis. Using Raman spectroscopy in combination with multivariate analysis, the spectral characteristics of the HC, SPC, MC, DCIS, and IDC samples were further extracted by PCA loading and score plots. We also confirmed that the tissue classification model based on the PCA–LDA algorithm, together with LOOCV, was able to distinguish the different breast tissue types. Therefore, this study illustrates the feasibility of Raman spectroscopy combined with multivariate analysis for the diagnosis of breast cancer.

## Figures and Tables

**Figure 1 molecules-26-00921-f001:**
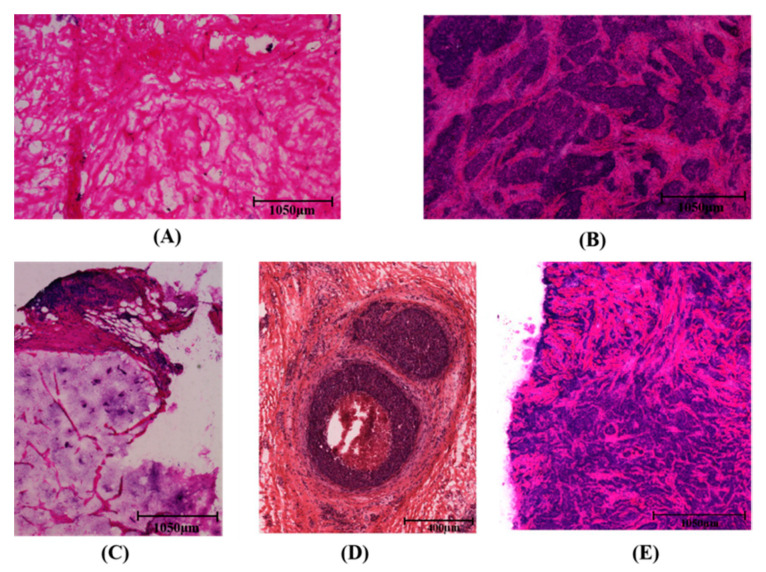
Hematoxylin and eosin (H&E) staining of samples of healthy breast tissue (**A**), solid papillary carcinoma (**B**), mucinous carcinoma (**C**), ductal carcinoma in situ (**D**), and invasive ductal carcinoma (**E**); scale bars: 1050 µm, 1050 µm, 1050 µm, 400 µm and 1050 µm, respectively.

**Figure 2 molecules-26-00921-f002:**
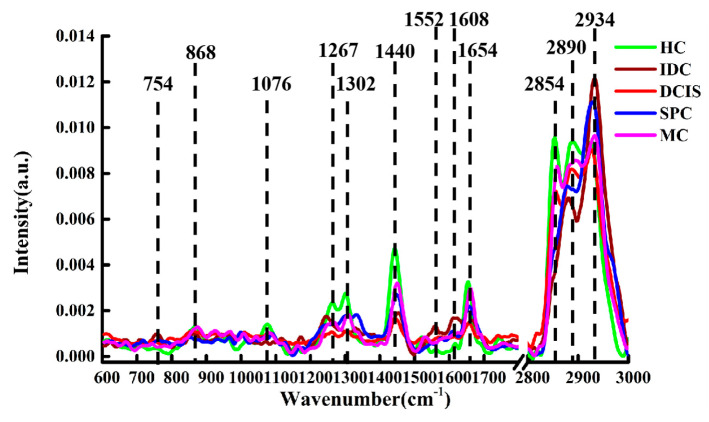
Normalized spectra of HC, IDC, DCIS, SPC, and MC tissue samples.

**Figure 3 molecules-26-00921-f003:**
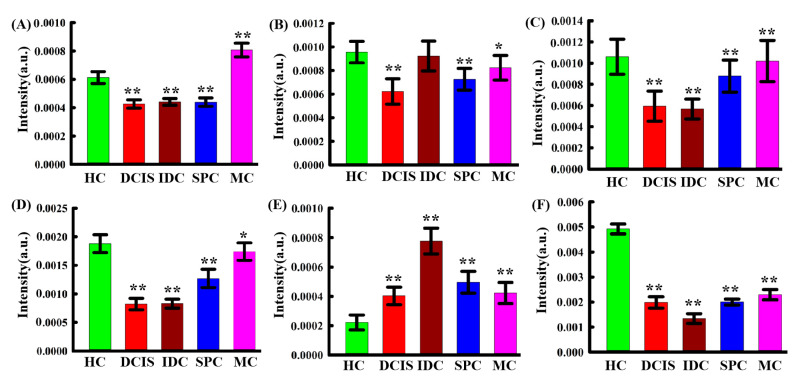
Bar graphs showing variations in the intensity of spectral peak corresponding to various biochemical components in HC, IDC, DCIS, SPC, and MC samples. The mean ± standard deviation is indicated. (**A**) Collagen (868 cm^−1^), (**B**) lipid (1267 cm^−1^), (**C**) lipid (1302 cm^−1^), (**D**) lipid (1440 cm^−1^), (**E**) protein (1608 cm^−1^), (**F**) lipid (2890 cm^−1^). Statistical signifificance was determined by one-way ANOVA followed by a Tukey’s HSD post-hoc test. Asterisks indicate levels of signifificance, * *p* < 0.05, ** *p* < 0.01.

**Figure 4 molecules-26-00921-f004:**
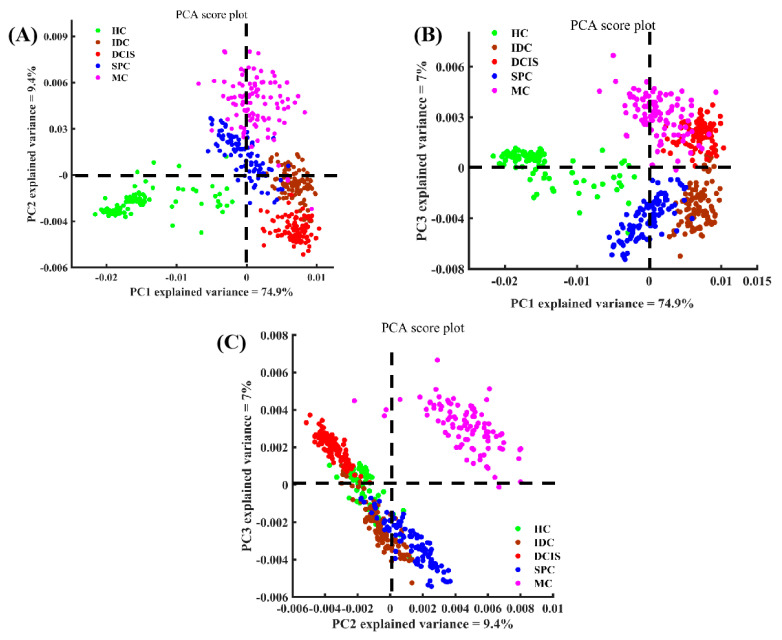
Scatter plots of the diagnostically significantly principal components scores for the different types of breast tissue. (**A**) PC1 versus PC2; (**B**) PC1 versus PC3; and (**C**) PC2 versus PC3.

**Figure 5 molecules-26-00921-f005:**
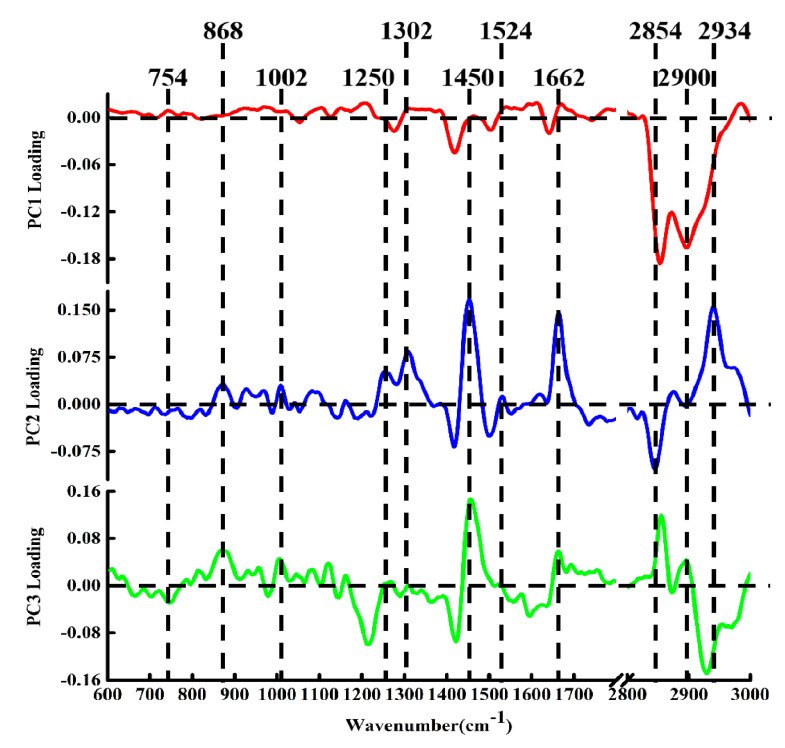
The PCA loading spectra of PC1, PC2, and PC3.

**Figure 6 molecules-26-00921-f006:**
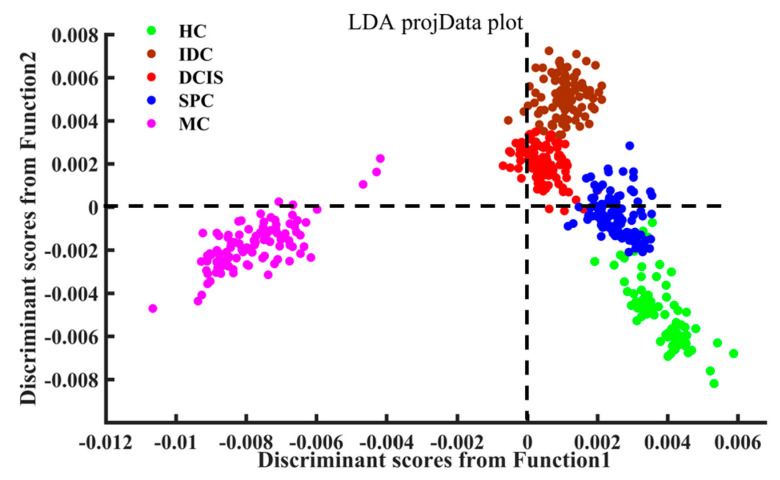
Scatter plot of linear discriminant scores for different types of tissue.

**Table 1 molecules-26-00921-t001:** Detailed information of the samples. HC, healthy control, SPC, solid papillary carcinoma, MC, mucinous carcinoma, IDC, invasive ductal carcinoma, DCIS, ductal carcinoma in situ.

Types of Cancer	Number of Samples	Number of Patients
HC	12	4
SPC	3	3
MC	3	3
IDC	6	6
DCIS	6	6

**Table 2 molecules-26-00921-t002:** LOOCV results of the PCA–LDA model. Actual: pathological diagnosis results. Predict: Raman spectra combined with the classification model provided the diagnostic results.

Actual/Predict	HC	IDC	DCIS	SPC	MC
HC	78	0	0	2	0
IDC	0	77	3	0	0
DCIS	0	0	80	0	0
SPC	0	0	0	80	0
MC	0	0	0	0	80

**Table 3 molecules-26-00921-t003:** Results of the PCA–LDA model on the test set. Actual: Pathological diagnosis results. Predict: Raman spectra combined with the classification model provided the diagnostic results.

Actual/Predict	HC	IDC	DCIS	SPC	MC
HC	20	0	0	0	0
IDC	0	20	0	0	0
DCIS	0	0	20	0	0
SPC	0	0	0	20	0
MC	0	0	0	0	20

## Data Availability

The data presented in this study are available on request from the corresponding author. The data are not publicly available due to sample.

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
