# Peer review of "Raman Microspectroscopic Investigation and Classification of Breast Cancer Pathological Characteristics"

_molecules, 2021, doi:10.3390/molecules26040921_

Round 1
Reviewer 1 Report
The study by Heping Li et al. reports data confirming that breast cancer features can be accurately identified by label-free Raman spectroscopy. In addition, authors demonstrated that their approach is able to accurately distinguish different breast cancer subtype on fresh frozen tissue.
Even if the study is ambitious and potentially interesting for the scientific community there are some major limitations that do not permit its publication in its current version. Here a few general considerations and below a more detailed comment.
First, the study aims are not clear and the results are not adequately discussed considering current unmet clinical needs. In other words, considering that Raman spectroscopy already demonstrated the possibility to distinguish tumour features in differ tissues (including breast), nowadays we “researchers in the Raman field” should better specify the novelty of current studies. I think authors should better detail the novelty reported, how these data may help the development of new diagnostic tools and/or how these data may help to better understand cancer biology. And this should be done also clearly describing how this experimental setup could be transferred to clinics and/or used to develop further diagnostic technologies.
Second, some technical and experimental details are missing or not sufficiently detailed.
- Page 1, line 41: In the “introduction” authors report that “However, it usually takes 1-2 weeks to evaluate the edge of the operation, and if the margin is positive, the patient needs to have a second operation [..]..”. This is true but author should also consider and mention that in the current practice also the intraoperative margin assessment is commonly performed thanks to the evaluation of fresh-frozen slices. This is important especially if we consider the compatibility of Raman approaches with fresh-frozen unstained samples.
- Page 2: in “introduction” authors mentioned several examples of Raman approaches applied to cancer detection and diagnosis. At the same time authors did not mention some relevant and recent advances in the specific field of breast cancer studied by Raman. For example I would suggest to mention recent studies by Ioan Nothinger (for example: Shipp, Dustin W., et al. "Intra-operative spectroscopic assessment of surgical margins during breast conserving surgery." Breast Cancer Research 20.1 (2018): 69) and by Renzo Vanna ( Vanna, R., Morasso, C., Marcinnò, B., Piccotti, F., Torti, E., Altamura, D., ... & Bunk, O. (2020). Raman Spectroscopy Reveals That Biochemical Composition of Breast Microcalcifications Correlates with Histopathologic Features. Cancer Research, 80(8), 1762-1772.)
- Page 2: in “introduction”, last paragraph, it is usually appreciated to clearly state the study aims, including the type of samples used (e.g. fresh, fixed, frozen etc.) and the proposed advances beyond the state of art if compared with the literature mentioned in the previous paragraphs. Otherwise it is not clear which is the novelty of the study and this should be very clear before to start reading Results and discussion, in particular because the combination of Raman and multivariate analysis on breast cancer assessment has already been reported.
- Page 2: I would use “HC” instead of “H” for healthy control samples.
- Page 2: the following sentence “ The form of cancer type can be ascertained by the infiltration degree of cancer cells represented by black particles in the H&E stained images” is not correct from the pathological point of view. The cancer type and the cancer infiltration is not represented by “black particles” but by more complex and detailed morphological changes described by pathologists. In any case, “black particles” is not a histological marker of disease. One could refer to cancer-cell infiltration that can be detected by darker H&E appearance due to the presence of numerous nuclei (but this is still a partial description) . In general, for what concern the pathological details, I would recommend to refer to official WHO guidelines (“Tavassoli FA, Devilee P: Pathology and Genetics: Tumours of the Breast and Female Genital Organs. WHO Classification of Tumours series - volume IV. Lyon, France: IARC Press; 2003. 250pp. ISBN 92 832 2412 4”) or, if not accessible, to similar open-access guidelines and documents. I would also suggest a detailed review by pathologist who provided the samples.
- Page 2: at the beginning of Results, before or after the pathological analysis, authors should clearly state the number of samples and patients (and corresponding pathology) included in the study, even if the same data were reported in “materials and methods”. Sometime a table (in the text or in the supplementary material) can help.
- Page 3: Authors start with “The normalized spectra of healthy and cancerous breast tissues” but is not easy to understand how they produced the data. Considering that in the format of Molecules journal “methods” comes after “results” , authors should better detail here the acquisition details (i.e. mapping vs single acquisition; how the regions was selected etc.). This is fundamental for a Raman-based study. In any case, even in material and methods, the acquisition strategy is not clearly reported. Which criteria were used to acquire spectra on a certain tissue region? Breast tissue morphology, especially in case of cancer, can be highly heterogeneous (connective tissue, fat tissue, cellular regions, calcifications, ducts, etc) can be observed in samples. Furthermore, authors say in Methods “500 spectra for each sample” but in figures and in table 1 it seems that 100 spectra were acquired for each sample.
- Page 3, Line 101: authors write “which may be related to the high rate of cell division and the thinning of the lipid cell membranes in the process of cancer cell invasion and migration”. This can be true but healthy breast is typically a fat-rich tissue and this composition is normally altered when cancer cells start to grow. Therefore, the data are compatible with the presence of cancer (low fat content) if compared with HC tissue (high fat content), but the interpretation of data is not completely reasonable. The thinning of lipid cell membrane can hardly been seen by Raman approaches. Similarly, in line 108 authors report: “it may be associated with the large quantity of protein synthesized by cancer cells during uncontrolled growth”. This can be true but most of protein contribute simply derive by the fact that higher cellularity (i.e. less lipids and more cells (= DNA and proteins)) is associated with cancer. In addition, these observations (presence of fat-rich molecules) underline the need to specify the regions selected for acquisition.
- Figure 2: it is not a fundamental comment but, for more clarity, it is normally appreciated if color-codes and sample order follow pathology. For example, the order of malignancy is HC, DCIS and then carcinomas. I would therefore use this order in the text and I figures, using an approprieate color-coding. For example, HC in green (meaning “ok”, DCIS in blu, and carcinoma in more reddish colour (meaning “disease…”). Just a suggestion.
- Figure 2: Phenylalanine signal is almost not visible as normally occurs in Raman data of tissues. Is this due to high-fluorescence background? May authors discuss this data?
- Figure 3: data reported in this figure are very interesting and well represent the biochemical changes occurring in cancer. For example, it is interesting to see that lipids are low in IDC but they significantly increase in MC, due to the presence of typical mucous and fatty-rich features. I think similar discussion should be also reported. Here authors have performed ANOVA and Tukey’s HSD but the results of these tests and statistical performances were not reported, and these are missing data. I would add in the plots significant values (e.g. p<xxx or asterisks or similar symbols) or numbers in the text, to show where data are significantly different.
- Page 5: data extracted by PCA are interesting and coherent to what observed in spectra. Authors say that PC2 largely represent MC samples and that is correct. At the same time I would suggest to better underline that the negative portion of PC2 well describe that MC show differences around 2854 1/cm. In particular it is clear (also from the spectra) that MC band in the CH region related to CH2 is not precisely centered around 2954 but that is red-shifted (around 2960 1/cm). It would be interesting to understand the assignation of these signals in the MC sample context.
- Page 6 and Page 8: authors should report the number of PCs used for LDA and the variance percentage represented by these PCs. Ideally these PCs should be reported as supplementary material.
- Page 8: authors should specify how they performed background subtraction (polynomial? Which degree?) and noise removal.
If authors will improve the manuscript also according to these suggestions, I think it could be accepted for publication in “Molecules”.
Author Response
The study by Heping Li et al. reports data confirming that breast cancer features can be accurately identified by label-free Raman spectroscopy. In addition, authors demonstrated that their approach is able to accuratelydistinguish differentbreast cancer subtype on fresh frozen tissue.
Even if the study is ambitious and potentially interesting for the scientific community there are some major limitationsthat do not permit its publication in its current version. Here
a few general considerations and below a more detailed comment.
First, the study aims are not clear and the results are not adequately discussed considering current unmet clinical needs.In other words, considering that Raman spectroscopy already demonstrated the possibility to distinguish tumour features in differ tissues (including breast), nowadays we “researchers in the Raman field” should better specify the novelty ofcurrent studies. I think authors should better detail the novelty reported, how these data may help the development ofnew diagnostic tools and/or how these data may help to better understand cancer biology. And this should be done also clearly describing how this experimental setup could be transferred to clinics and/or used to develop further diagnostic technologies.
Second, some technical and experimental details are missing or not sufficiently detailed.
Page 1, line 41: In the “introduction” authors report that “However, it usually takes 1-2 weeks to evaluate the edge of the operation, and if the margin is positive, the patient
needs to have a second operation [..]..”. This is true but author should also consider and mention that in the currentpractice also the intraoperative margin assessment is commonly performed thanks to the evaluation of fresh-frozen slices. This is important especially if we consider the compatibility of Raman approaches with fresh-frozen unstained samples.
Answers:Thank you for your comments, we have made a supplement in the article.
Page 2: in “introduction” authors mentioned several examples of Raman approaches applied to cancer detection anddiagnosis. At the same time authors did not mention some relevant and recent advances in the specific field of breastcancer studied by Raman. For example Iwould suggest to mention recent studies by Ioan Nothinger(for example: Shipp,Dustin W., et al. "Intra-operative spectroscopic assessment of surgical margins during breast conserving surgery." BreastCancer Research 20.1(2018): 69) and by Renzo Vanna ( Vanna, R., Morasso, C.,Marcinnò, B., Piccotti, F., Torti, E.,Altamura, D., ... & Bunk,O. (2020). Raman Spectroscopy Reveals That BiochemicalComposition of BreastMicrocalcifications Correlates with Histopathologic Features. Cancer Research, 80(8), 1762-1772.)
Answers: Thank you for your comments. After carefully reading your suggested article, we have made corresponding supplements.
Page 2: in “introduction”, last paragraph, it is usually appreciated to clearly state the study aims, including the type ofsamples used (e.g. fresh, fixed, frozen etc.) and the proposed advances beyond the state of art if compared with the literature mentioned in the previous paragraphs. Otherwise it is not clear which is the novelty of the study and this should be very clear before to start reading Results and discussion, in particular because the combination of Raman andmultivariate analysis on breast cancer assessment has already been reported.
Answers: Thank you for your comments, we have made the revision according to your suggestion.
Page 2: I would use “HC” instead of “H” for healthy control samples.
Answers: Thank you for your suggestion. We have done the modification accordingly.
Page 2: the following sentence “ The form of cancer type can be ascertained by the infiltration degree of cancer cellsrepresented by black particles in the H&E stained images” is not correct from the pathological point of view. The cancertype and the cancer infiltration is not represented by “black particles” but by more complex and detailed morphologicalchanges described by pathologists. In any case, “black particles” is not a histological marker of disease. One could referto cancer-cell infiltration that can be detected by darker H&E appearance due to the presence of numerous nuclei (butthis is still a partial description). In general, for what concern the pathological details, I would recommend to refer toofficial WHO guidelines (“Tavassoli FA, Devilee P: Pathology and Genetics: Tumours of the Breast and Female GenitalOrgans. WHO Classification of Tumours series - volume IV. Lyon, France: IARC Press; 2003. 250pp. ISBN 92 832 24124”) or, if not accessible, to similar open-access guidelines and documents. I would also suggest a detailed review bypathologist who provided the samples.
Answers:Thank you for your suggestion. We have done the modification accordingly.
Page 2: at the beginning of Results, before or after the pathological analysis, authors should clearly state the number ofsamples and patients (and corresponding pathology) included in the study, even if the same data were reported in“materials and methods”. Sometime a table (in the text or in the supplementary material) can help.
Answers:Thank you for your comments, we have made a supplement in the article, as shown in Table 1.
Page 3: Authors start with “The normalized spectra of healthy and cancerous breast tissues” but is not easy to understand how they produced the data. Considering that in the format of Molecules journal “methods” comes after “results” ,authors should better detail here the acquisition details (i.e. mapping vs single acquisition; how the regions was selected etc.). This is fundamental for a Raman-based study. In any case, even in material and methods, the acquisition strategy isnot clearly reported. Which criteria were used to acquire spectra on a certain tissue region?Breast tissue morphology,especially in case of cancer, can be highly heterogeneous (connective tissue, fat tissue, cellular regions, calcifications,ducts, etc) can be observed in samples. Furthermore, authors say in Methods “500 spectra for each sample” but in figures and in table 1 it seems that 100 spectra were acquired for each sample.
Answers:Raman spectra were acquired using an Alpha 500R confocal Raman microscopy system (WITec GmbH, Germany) coupled with a helium neon (He-Ne) continuous 633 nm laser (35 mW at 633 nm, Research Electro-Optics, Inc., USA). A 3.5 μm diameter single-mode optical fiber was used to couple the laser radiation into an upright microscope. After passing through a holographic 633 nm bandpass fifilter, the excitation laser beam was collimated into a 100×objective lens (NA = 1.25, EC Epiplan-Neoflfluar, Zeiss, Germany).The sample was placed on a multi-axis piezo scanning stage (P-524K081, PI GmbH, Germany).The Raman photons were collected by the same objective lens, and then sent into a multimode optical fifiber (50 μm diameter) through a holographic edge fifilter. The spectral signal was monitored by a spectrometer (UHTS300, WITec GmbH, Germany) incorporating a 600 mm−1 grating blazed at 500 nm with a resolution of approximately 3 cm−1 over the range 0 to 3000 cm−1 . The Raman spectrum was recorded by a back illuminated deep-depletion charge coupled device (CCD) camera (Du401A-BR-DD-352, Andor Technology, UK) working at −60 °C. Each spectrum was recorded over a period of 1.5s. The area of spectrum collection is determined by pathological experts, and 100 single spectra are collected for each tissue type (We have done the modification accordingly.).
Page 3, Line 101: authors write “which may be related to the high rate of cell division and the thinning of the lipid cell membranes in the process of cancer cell invasion and migration”. This can be true but healthy breast is typically a fat-rich tissue and this composition is normally altered when cancer cells start to grow. Therefore, the data are compatible with the presence of cancer (low fat content) if compared with HC tissue (high fat content), but the interpretation of data is not completely reasonable. The thinning of lipid cell membrane can hardly been seen by Raman approaches. Similarly, in line 108 authors report: “it may be associated with the large quantity of protein synthesized by cancer cells during uncontrolled growth”.This can be true but most of protein contribute simply derive by the fact that higher cellularity (i.e. less lipids and more cells (= DNA and proteins)) is associated with cancer. In addition, these observations (presence of fat-rich molecules) underline the need to specify the regions selected for acquisition.
Answers: Thank you for your comments. I quite agree with you, “the thinning of lipid cell membrane can hardly been seen by Raman approaches.” However, in Raman spectra, the decrease of lipid intensity reflects the decrease of lipid content in tissues, which may be related to the high rate of cell division and the thinning of the lipid cell membranes in the process of cancer cell invasion and migration. We have done the modification accordingly.The results of this study are similar to those of Tanmoy Bhattacharjee et al. “In vivo Raman spectroscopy of breast tumors prephotodynamic and postphotodynamic therapy”. Journal of Raman Spectroscopy 2018, 49, (5), 786-791.
Figure 2: it is not a fundamental comment but, for more clarity, it is normally appreciated if color-codes and sample order follow pathology. For example, the order of malignancy is HC, DCIS and then carcinomas. I would therefore use this order in the text and I figures, using an approprieate color-coding. For example, HC in green (meaning “ok”, DCIS in blu, and carcinoma in more reddish colour (meaning “disease…”). Just a suggestion.
Answers: Thank you for your comments, we have made the revision according to your suggestion.
Figure 2: Phenylalanine signal is almost not visible as normally occurs in Raman data of tissues. Is this due to high-fluorescence background? May authors discuss this data?
Answers: This may not have much to do with the high fluorescence background, this may be related to the low content of phenylalanine in breast tissue. The results of this paper are similar to those of other research groups:
- Han B , Du Y , Fu T , et al. Differences and Relationships Between Normal and Atypical Ductal Hyperplasia, Ductal Carcinoma In Situ, and Invasive Ductal Carcinoma Tissues in the Breast Based on Raman Spectroscopy[J]. Applied Spectroscopy, 2017, 71(2):300-307.
- Li Q , Gao Q , Zhang G . Classification for breast cancer diagnosis with Raman spectroscopy[J]. Biomedical Optics Express, 2014, 5(7):2435.
- Haka A , Shafer-Peltier K , Fitzmaurice M , et al. Diagnosing breast cancer by using Raman spectroscopy[J]. Proceedings of the National Academy of Sciences of the United States of America, 2005, 102(35):p. 12371-12376.
Figure 3: data reported in this figure are very interesting and well represent the biochemical changes occurring in cancer. For example, it is interesting to see that lipids are low in IDC but they significantly increase in MC, due to the presence of typical mucous and fatty-rich features. I think similar discussion should be also reported. Here authors have performed ANOVA and Tukey’s HSD but the results of these tests and statistical performances were not reported, and these are missing data. I would add in the plots significant values (e.g. p<xxx or asterisks or similar symbols) or numbers in the text, to show where data are significantly different.
Answers: Thank you for your comments, we have made the revision according to your suggestion.
Page 5: data extracted by PCA are interesting and coherent to what observed in spectra. Authors say that PC2 largely represent MC samples and that is correct. At the same time I would suggest to better underline that the negative portion of PC2 well describe that MC show differences around 2854 1/cm. In particular it is clear (also from the spectra) that MC band in the CH region related to CH2 is not precisely centered around 2954 but that is redshifted (around 2960 1/cm). It would be interesting to understand the assignation of these signals in the MC sample context.
Answers: Thank you for your comments, we have done the modification accordingly, and the biochemical distribution of the peak was indicated.
Page 6 and Page 8: authors should report the number of PCs used for LDA and the variance percentage represented by these PCs. Ideally these PCs should be reported as supplementary material.
Answers: The first PC accounted for the largest variance within the spectral dataset (PC1, 74.9%), while PC2 and PC3 represented 9.4% and 7% of the total variance, respectively. The first three PC scores were used as the input variables for LDA to establish the final diagnostic model.
Page 8: authors should specify how they performed background subtraction (polynomial? Which degree?) and noise removal.
Answers: The background subtraction including a nine-order polynomial fit to subtract the spectral background and a five-order Savitzky-Golay smoothing to noise removal.
If authors will improve the manuscript also according to these suggestions, I think it could be accepted for publication in “Molecules”.

Reviewer 2 Report
The authors aimed to improve the capability of Raman spectroscopic technique for clinical application in breast cancer diagnosis, and demonstrated Raman measurement with a confocal Raman microscopy incorporating multivariate analysis. Overall, the manuscript is well-written, and the experimental results are clearly presented. However, several questions and concerns which should be corrected prior to publish arise as listed below;
1. Page 1, line 20. "raman" should be "Raman".
2. Page 3, Figure 2. Is each spectrum representative data obtained from the patients? Or, averaged spectra in each histological category? The data quality is relatively good for the short exposure time of 1.5 seconds per spectrum. Please describe the experimental condition for spectral acquisition in Fig.2.
3. Page 8, Line24. "500 spectra were randomly acquired..." How many samples were obtained from each patient? How many sections were measured in each samples? How many point were measured in each section? Please clarify these points.
4. The authors conclude that the spectral characteristics in the H, SPC, MC, DCIS and IDC were distinguishable by Raman spectroscopy combined with PCA-LDA algorithm. However, the relationship between the histopathological findings and the alterations in Raman peaks are still unclear. If possible, please add the discussion about this, otherwise please add the limitations of this study regarding the clinical practice.
5. Page 8, Line 27. Supplementary material is not found.
6. Page 9, Line 27. Please mention the funding.
Author Response
The authors aimed to improve the capability of Raman spectroscopic technique for clinical application in breast cancerdiagnosis, and demonstrated Raman measurement with a confocal Raman microscopy incorporating multivariate analysis. Overall, the manuscript is well-written, and the experimental results are clearly presented. However, severalquestions and concerns which should be corrected prior to publish arise as listed below;
- Page 1, line 20. "raman" should be "Raman".
Answers: We have done the modification accordingly.
- Page 3, Figure 2. Is each spectrum representative data obtained from the patients? Or, averaged spectra in eachhistological category? The data quality is relatively good for the short exposure time of 5 seconds per spectrum. Please describe the experimental condition for spectral acquisition in Fig.2.
Answers: Raman spectra were acquired using an Alpha 500R confocal Raman microscopy system (WITec GmbH, Germany) coupled with a helium neon (He-Ne) continuous 633 nm laser (35 mW at 633 nm, Research Electro-Optics, Inc., USA). A 3.5 μm diameter single-mode optical fiber was used to couple the laser radiation into an upright microscope. After passing through a holographic 633 nm bandpass fifilter, the excitation laser beam was collimated into a 100×objective lens (NA = 1.25, EC Epiplan-Neoflfluar, Zeiss, Germany).The sample was placed on a multi-axis piezo scanning stage (P-524K081, PI GmbH, Germany).The Raman photons were collected by the same objective lens, and then sent into a multimode optical fifiber (50 μm diameter) through a holographic edge fifilter. The spectral signal was monitored by a spectrometer (UHTS300, WITec GmbH, Germany) incorporating a 600 mm−1 grating blazed at 500 nm with a resolution of approximately 3 cm−1 over the range 0 to 3000 cm−1 . The Raman spectrum was recorded by a back illuminated deep-depletion charge coupled device (CCD) camera (Du401A-BR-DD-352, Andor Technology, UK) working at −60 °C. In total, 100 spectra were randomly acquired from each sample (H, DCIS, SPC, MC and IDC), respectively. Each spectrum was recorded over a period of 1.5s. The spectra of each tissue type were averaged and then normalized for comparison.
- Page 8, "500 spectra were randomly acquired..." How many samples were obtained from each patient? How many sections were measured in each samples? How many point were measured in each section? Please clarify these points.
Answers:The number of sample and patients are shown in Table 1. Three sections were taken from each sample, and 20-30 points were randomly measured from each section. Finally, 100 spectra of each tissue type were randomly selected for analysis.
- The authors conclude that the spectral characteristics in the H, SPC, MC, DCIS and IDC were distinguishable byRaman spectroscopy combined with PCA-LDA algorithm. However, the relationship between the histopathologicalfindings and the alterations in Raman peaks are still unclear. If possible, please add the discussion about this, otherwise please add the limitations of this study regarding the clinical practice.
Answers: Thank you for your comments. We add the limitations of Raman spectroscopy in clinical application to the conclusion.
- Page 8, Line 27. Supplementary material is not
Answers: There is no supplementary material for this study.
- Page 9, Line 27. Please mention the funding.
Answers: Thank you for your comments. We have made a supplement in this paper.

Reviewer 3 Report
In the presented work, Li et al. describe a workflow to perform a statistical classification of healthy breast tissue and four breast cancer tissue types using Raman spectroscopy-based models. The results show that the presented model, based on a combination of feature extraction (PCA) and linear classifier (LDA), can discriminate the five tissue types with very high accuracy.
However, the work lacks the novelty to be considered of interest for the public. In the last 10-15 years, several works have already proved the value of Raman spectroscopy as a technological platform capable of extracting useful information to discriminate various cancer types.
In particular, works like Haka et al. "Diagnosing breast cancer by using Raman spectroscopy." Proceedings of the National Academy of Sciences (2005) have already described Raman spectroscopy's application to discriminate different breast cancer tissue types.
The biochemical insight provided in the article is based on a speculative interpretation of the detected informative spectroscopic features, with no evidence provided. For this reason, it gives little additional value to the work.
Unfortunately, the presented work does not provide any additional knowledge to reach a sufficient level of novelty. For this reason, I cannot recommend the publication on 'Molecules'.
Comments:
- Line 60 "especially for clinicians or biomedical researchers without a solid background in 60 physics": there is no evidence to support this general statement.
- PCA-LDA: it is not clear how the modelling was performed. From the text, it seems that the authors performed PCA on the entire dataset. Therefore, they selected the number of principal components based on a univariate test (ANOVA) and finally used the selected scores to perform the classification with an LDA model. How was the LOOCV procedure performed? Did the authors calculate the PCA scores from the entire dataset followed by a LOOCV using the scores as features for the LDA model? Or did they calculate the scores from the only training set of each round of the LOOCV? In that case, how many principal components were found to be informative in each round of the LOOCV?
Author Response
In the presented work, Li et al. describe a workflow to perform a statistical classification of healthy breast tissue and fourbreast cancer tissue types using Raman spectroscopy-based models. The results show that the presented model, based ona combination of feature extraction (PCA) and linear classifier (LDA), can discriminate the five tissue types with very high accuracy.
However, the work lacks the novelty to be considered of interest for the public. In the last 10-15 years, several workshave already proved the value of Raman spectroscopy as a technological platform capable of extracting usefulinformation to discriminate various cancer
types.
In particular, works like Haka et al. "Diagnosing breast cancer by using Raman spectroscopy." Proceedings of theNational Academy of Sciences (2005) have already described Raman spectroscopy's application to discriminate differentbreast cancer tissue types.The biochemical insight provided in the article is based on a speculative interpretation of the detected informative
spectroscopic features, with no evidence provided. For thisreason, it gives little additional value to the work.
Unfortunately, the presented work does not provide any additional knowledge to reach a sufficient level of novelty. Forthis reason, I cannot recommend the publication on'Molecules'.
Comments:
Line 60 "especially for clinicians or biomedical researchers without a solid background in 60 physics": there is noevidence to support this general statement.
Answers: Thanks for your advice. We revised this sentence as follows: Even though, it is still necessary to building an correlation between the spectra with tissue pathological state by using multivairat analysis algorithm for helping clinican make for helping clinicians to make precise diagnosis, which might be more appropriate.
PCA-LDA: it is not clear how the modelling was performed. From the text, it seems that the authors performed PCA onthe entire dataset. Therefore, they selected the number of principal components based on a univariate test (ANOVA) andfinally used the selected scores to perform the classification with an LDA model. How was the LOOCV procedure performed? Did the authors calculate the PCA scores from the entire dataset followed by a LOOCV using the scores asfeatures for the LDA model? Or did they calculate the scores from the only training set of each round of the LOOCV? Inthat case, how many principal components were found to be informative in each round of the LOOCV?
Answers: Each pre-processed spectrum was normalized by the integrated area under the curve to eliminate interference from unrelated factors and provide a superior comparison of the spectral features of different tissues. Subsequently, the entire dataset containing all spectra were mean-centered and used for PCA and LDA in multivariate analysis. One-way ANOVA was used to identify the most diagnostically significant PCs (P < 0.01) to better classify the tissue spectral. Subsequently, the PC scores were used as input for LDA to generate diagnostic algorithms. The performance of the PCA-LDA diagnostic algorithm was validated in an unbiased manner using leave-one-out cross-validation (LOOCV) method.
In the LOOCV validation procedure, one tissue spectrum was left out and the PCA-LDA modeling was redeveloped using the remaining Raman spectra. The redeveloped PCA-LDA diagnostic model was then used to classify the withheld Raman spectra. This process was repeated iteratively until all withheld Raman spectra were classified.
We calculate the PCA scores for the entire dataset and then use the significant PCs scores as the features of the LDA model for LOOCV validation.

Round 2
Reviewer 1 Report
Here I will detail my comments considering answers from authors.
In general, even if the journal requires a really small (too small) number of days for the answers, I would strongly suggest to authors to answer to comments by reviewers detailing which parts (or new text) has been added and were it was implemented in the text, and to be more careful in implementing the text. Othervise it is not easy to understand how the article has been improved.
Authors implemented most of my suggestions but I still have a few minor but important modifications that authors should add before eventual publications:
- in the previous version I did not notice that authors written “it has been widely accepted by clinicians and research communities for the early diagnosis of cancer”. It is not true that RS has been “accepted by clinicians…for the diagnosis of cancer” (only one FDA approved approach on Skin cancer has been approved). I would suggest to modify this sentence just reporting that the results are promising and so on.
- even if I suggested to avoid using “black particle” when referring to cancer cell infiltration this wrong nomenclature is still there. Cancer cells are not “particles” and are not “black”. “particles” is normally referred to exogenous material. If black stain is present on tissue sample this is due to artifacts. Cell nuclei are stained by hematoxilin which has a typical purple color.
- in my previous report I asked to “better detail here the acquisition details (i.e. mapping vs single acquisition; how the regions was selected etc.)” at the beginning of Results to immediately state which is the used approach. To specify this in “methods” is not enough, especially if “methods” comes after “results”.
- in addition I’ve also asked to state “Which criteria were used to acquire spectra on a certain tissue region”. Authors answered here (by pathologist help) but the full text seems not reporting this fundamental information.
- I was suggesting to change the order of samples according to pathology (HC, DCIS and then carcinomas (IDC etc.)) but authors just changed the colors. That is not a relevant problem for the scientific point of view but, as I said above, I would have appreciated more attention to reviewer comments. I hope this can help for next articles.
Author Response
Here I will detail my comments considering answers from authors.
In general, even if the journal requires a really small (too small) number of days for the answers, I would strongly suggest to authors to answer to comments by reviewers detailing which parts (or new text) has been added and were it was implemented in the text, and to be more careful in implementing the text. Othervise it is not easy to understand how the article has been improved.
Authors implemented most of my suggestions but I still have a few minor but important modifications that authors should add before eventual publications:
- in the previous version I did not notice that authors written “it has been widely accepted by clinicians and research communities for the early diagnosis of cancer”. It is not true that RS has been “accepted by clinicians…for the diagnosis of cancer” (only one FDA approved approach on Skin cancer has been approved). I would suggest to modify this sentence just reporting that the results are promising and so on.
Answers: Thank you for your suggestion. We have revised it as follows:
After years of development, clinicians and researcher communities believe that RS is promising for early diagnosis of cancer, the identification of cancer progression and in-traoperative guidance
- even if I suggested to avoid using “black particle” when referring to cancer cell infiltration this wrong nomenclature is still there. Cancer cells are not “particles” and are not “black”. “particles” is normally referred to exogenous material. If black stain is present on tissue sample this is due to artifacts. Cell nuclei are stained by hematoxilin which has a typical purple color.
Answers: Thank you for your comments. We have made some modifications like:
In the process of cancer-cell infiltration, a large number of nuclei can be detected by purple H&E appearance.
- in my previous report I asked to “better detail here the acquisition details (i.e. mapping vs single acquisition; how the regions was selected etc.)” at the beginning of Results to immediately state which is the used approach. To specify this in “methods” is not enough, especially if “methods” comes after “results”.
Answers: Thank you for your comments. At the beginning of the results, we have added the details of the spectrum acquisition.
- in addition I’ve also asked to state “Which criteria were used to acquire spectra on a certain tissue region”. Authors answered here (by pathologist help) but the full text seems not reporting this fundamental information.
Answers: At the beginning of the result, we have added it. Raman spectra were acquired using an Alpha 500R confocal Raman microscopy system (WITec GmbH, Germany) coupled with a helium neon (He-Ne) continuous 633 nm laser (35 mW at 633 nm, Research Electro-Optics, Inc., USA). Each spectrum was recorded over a period of 1.5s. The area of spectrum collection is determined by pathological experts, and 100 single spectra are collected for each tissue type. The spectra of each tissue type were averaged and then normalized for comparison.
- I was suggesting to change the order of samples according to pathology (HC, DCIS and then carcinomas (IDC etc.)) but authors just changed the colors. That is not a relevant problem for the scientific point of view but, as I said above, I would have appreciated more attention to reviewer comments. I hope this can help for next articles.
Answers: Thank you for your suggestion. In Figure 3, We have revised the order of samples according to your opinion.
Reviewer 2 Report
The authors have addressed all of my comments.
Author Response
Thanks for all your comments.
Reviewer 3 Report
I would like to thank the authors for replying to most of the concerns expressed in the previous review.
Unfortunately, due to the expressed concern about the novelty of the work, I cannot still recommend the article for a publication.
Additionally, the authors describe the statistical modelling approach as following:
- A PCA is performed on the entire dataset (identifying the number of informative components using an ANOVA significance test against the samples classes)
- The selected PCA scores (calculated from the entire dataset) are used as features for a leave-one-out cross-validation (LOOCV) to estimate the predictive performance of the model.
Unfortunately, the usage of the entire dataset to estimate the identify and estimate the PCA scores invalidates the LOOCV procedure.
The number of principal components and the corresponding scores are calculated using the entire dataset, thus also using the information coming from the samples that will be afterwards used as a test in each round of the cross-validation, with a consequent leakage of information.
The correct way of performing the LOOCV would be, in each round of the CV:
- Split the dataset in training and test sets
- Estimate the number of principal components from the only training set samples
- Calculate the PCA scores of the training set, then calculate the PCA scores of the test sample projecting it onto the training PCA loadings
- Fit an LDA model on the training PCA scores and predict the class of the test sample from its PCA scores
Instead, when using the entire dataset to calculate PCA:
- The number of informative components is estimated using also the information (spectrum and label) from the sample that will be used as a test
- The PCA scores use the information coming from the eventual sample that will be used as a test, then they are clearly informative about the test sample.
In this way, the performance of the model are likely to be overoptimistic, since the features used to predict the eventual test sample label are calculated using information coming from that test sample too.
Author Response
Answers: One of the purposes of PCA analysis of the entire dataset is to reduce the data dimensions and simplify the complexity of the dataset. Secondly, in order to extract significant features from the spectral dataset for in-depth analysis and study the spectral differences between different types of tissues. The following research group is consistent with us:
- Teh S K , Zheng W , Ho K Y , et al. Near-infrared Raman spectroscopy for early diagnosis and typing of adenocarcinoma in the stomach[J]. British Journal of Surgery, 2010, 97(4):550-557.
- Li Y , Pan J , Chen G , et al. Micro-Raman spectroscopy study of cancerous and normal nasopharyngeal tissues[J]. Journal of Biomedical Optics, 2013, 18(2):27003.
- Chen L , Wang Y , Liu N , et al. Near-infrared confocal micro-Raman spectroscopy combined with PCA–LDA multivariate analysis for detection of esophageal cancer[J]. Laser Physics, 2013, 23(6):065601.
During cross validation, we re-executed the LOOCV according to your suggestion, and the results are shown in the Table:
|
Actual\Predict |
HC |
IDC |
DCIS |
SPC |
MC |
|
HC |
96 |
0 |
0 |
4 |
0 |
|
IDC |
0 |
96 |
4 |
0 |
0 |
|
DCIS |
0 |
0 |
100 |
0 |
0 |
|
SPC |
0 |
0 |
0 |
100 |
0 |
|
MC |
0 |
0 |
0 |
0 |
100 |
The classification results in the above Table were compared with the original results, and it was found that there was no over-fitting of the model.
Round 3
Reviewer 3 Report
I would like to thank the authors for providing detailed answers to my comments.
My concerns about novelty are not solved.
Answers: One of the purposes of PCA analysis of the entire dataset is to reduce the data dimensions and simplify the complexity of the dataset. Secondly, in order to extract significant features from the spectral dataset for in-depth analysis and study the spectral differences between different types of tissues. The following research group is consistent with us:
- Teh S K , Zheng W , Ho K Y , et al. Near-infrared Raman spectroscopy for early diagnosis and typing of adenocarcinoma in the stomach[J]. British Journal of Surgery, 2010, 97(4):550-557.
- Li Y , Pan J , Chen G , et al. Micro-Raman spectroscopy study of cancerous and normal nasopharyngeal tissues[J]. Journal of Biomedical Optics, 2013, 18(2):27003.
- Chen L , Wang Y , Liu N , et al. Near-infrared confocal micro-Raman spectroscopy combined with PCA–LDA multivariate analysis for detection of esophageal cancer[J]. Laser Physics, 2013, 23(6):065601.
The fact that some previous works use a non-optimal statistical modelling approach does not solve the issue that I have raised.
As well known, PCA dimensionality reduction is based on its capability to identify directions along which the variance is maximized. When all samples are used to estimate the principal components, the test samples give an informative contribution to the estimation of the covariance matrix and the consequent linear space onto which the data points will be projected. As a consequence, a leakage of information occurs regarding the following predictive modelling.
The cross-validation scheme aims at simulating the scenario where one has a certain dataset (training set) used to fit a model and receives an unseen and unknown new sample to predict. The label of the test sample cannot be used in any step of the model fitting procedure. In your work, you are using the knowledge about the label of the test sample to select the number of informative principal components. How would you apply your procedure for a new sample with an unknown label (that the model is supposed to predict)?
During cross validation, we re-executed the LOOCV according to your suggestion, and the results are shown in the Table:
|
Actual\Predict |
HC |
IDC |
DCIS |
SPC |
MC |
|
HC |
96 |
0 |
0 |
4 |
0 |
|
IDC |
0 |
96 |
4 |
0 |
0 |
|
DCIS |
0 |
0 |
100 |
0 |
0 |
|
SPC |
0 |
0 |
0 |
100 |
0 |
|
MC |
0 |
0 |
0 |
0 |
100 |
The classification results in the above Table were compared with the original results, and it was found that there was no over-fitting of the model.
The predictive power is now 0.987. This is the correct performance of the model. The fact that the performances are similar does not justify the usage of an incorrect procedure.
Author Response
Comments and Suggestions for Authors I would like to thank the authors for providing detailed answers to my comments. My concerns about novelty are not solved. Answers: Thank you for your busy schedule to review my paper, and put forward valuable opinions. We carefully read the article which you recommended and found that Haka et al. mainly used Raman spectroscopy to study the normal breast tissue, benign lesions and cancerous tissue on the duct. In this paper, we mainly use Raman spectroscopy combined with multivariate analysis to study the healthy breast tissue, ductal carcinoma in situ, invasive ductal carcinoma, mucinous carcinoma and solid papillary carcinoma. This paper not only introduces the biochemical composition of different breast tissues, but also establishes a new diagnostic model. In addition, compared with the study of Haka et al., the biochemical composition and pathological characteristics of mucinous carcinoma and solid papillary carcinoma were additionally reported, which expanded the database of breast tissue. Secondly, we also reported the biochemical and histopathological changes in the classic pathological process of healthy tissue, ductal carcinoma in situ and invasive ductal carcinoma, which laid a solid foundation for the Raman spectroscopy image research of ductal carcinoma in situ and invasive ductal carcinoma. Answers: One of the purposes of PCA analysis of the entire dataset is to reduce the data dimensions and simplify the complexity of the dataset. Secondly, in order to extract significant features from the spectral dataset for in-depth analysis and study the spectral differences between different types of tissues. The following research group is consistent with us: Teh S K , Zheng W , Ho K Y , et al. Near-infrared Raman spectroscopy for early diagnosis and typing of adenocarcinoma in the stomach[J]. British Journal of Surgery, 2010, 97(4):550- 557. Li Y , Pan J , Chen G , et al. Micro-Raman spectroscopy study of cancerous and normal nasopharyngeal tissues[J]. Journal of Biomedical Optics, 2013, 18(2):27003. Chen L , Wang Y , Liu N , et al. Near-infrared confocal microRaman spectroscopy combined with PCA–LDA multivariate analysis for detection of esophageal cancer[J]. Laser Physics, 2013, 23(6):065601. The fact that some previous works use a non-optimal statistical modelling approach does not solve the issue that I have raised. As well known, PCA dimensionality reduction is based on its capability to identify directions along which the variance is maximized. When all samples are used to estimate the principal components, the test samples give an informative contribution to the estimation of the covariance matrix and the consequent linear space onto which the data points will be projected. As a consequence, a leakage of information occurs regarding the following predictive modelling. The cross-validation scheme aims at simulating the scenario where one has a certain dataset (training set) used to fit a model and receives an unseen and unknown new sample to predict. The label of the test sample cannot be used in any step of the model fitting procedure. In your work, you are using the knowledge about the label of the test sample to select the number of informative principal components. How would you apply your procedure for a new sample with an unknown label (that the model is supposed to predict)? During cross validation, we re-executed the LOOCV according to your suggestion, and the results are shown in the Table: Actual\Predict HC IDC DCIS SPC MC HC 96 0 0 4 0 IDC 0 96 4 0 0 DCIS 0 0 100 0 0 SPC 0 0 0 100 0 MC 0 0 0 0 100 The classification results in the above Table were compared with the original results, and it was found that there was no overfitting of the model. The predictive power is now 0.987. This is the correct performance of the model. The fact that the performances are similar does not justify the usage of an incorrect procedure. Answrs: Thank you for your comments. Your suggestions are provided with great help to our work. We have revised the selection of the number of principal components and the cross-validation process according to your suggestions. First, the initial dataset is divided into training set(80%) and test set(20%)according to a certain proportion. The training set is used to build the classifier, whereas the test set is used to evaluate the model classification performance. And use cross-validation to optimize model parameters (number of principal components) on the training set. This is made by first removing a certain number of samples from the training set and then building the model with the remaining samples, where the removed samples are predicted as a temporary validation set. This is performed for a certain number of repetitions usually until all training samples are excluded once from the training set and predicted as an external validation set. After determining the optimal number of principal components, leave-one-out cross-validation (LOOCV) is used to further verify the model, and the test set samples are used for evaluate the model classification performance. The results obtained are shown in the following table. The overall accuracy of cross-validation is 98.75%. The overall accuracy of the PCA-LDA model on the test set is 100%. Table 1. The LOOCV result of the PCA-LDA model. Actual: Pathological diagnosis results. Predict: Raman spectra combined with the classification model obtained the discriminant results. Actual\Predict HC IDC DCIS SPC MC HC 78 0 0 2 0 IDC 0 77 3 0 0 DCIS 0 0 80 0 0 SPC 0 0 0 80 0 MC 0 0 0 0 80 Table 2 . The results of the PCA-LDA model on the test set. Actual\Predict HC IDC DCIS SPC MC HC 20 0 0 0 0 IDC 0 20 0 0 0 DCIS 0 0 20 0 0 SPC 0 0 0 20 0 MC 0 0 0 0 20